# Topological in vitro loading of the budding yeast cohesin ring onto DNA

Masashi Minamino, Torahiko L Higashi, Céline Bouchoux, Frank Uhlmann

The ring-shaped chromosomal cohesin complex holds sister chromatids together by topological embrace, a prerequisite for accurate chromosome segregation. Cohesin plays additional roles in genome organization, transcriptional regulation, and DNA repair. The cohesin ring includes an ABC family ATPase, but the molecular mechanism by which the ATPase contributes to cohesin function is not yet understood. In this study, we have purified budding yeast cohesin, as well as its Scc2–Scc4 cohesin loader complex, and biochemically reconstituted ATP-dependent topological cohesin loading onto DNA. Our results reproduce previous observations obtained using fission yeast cohesin, thereby establishing conserved aspects of cohesin behavior. Unexpectedly, we find that nonhydrolyzable ATP ground state mimetics ADP·BeF$_2$, ADP·BeF$_3^-$, and ADP·AlF$_x$, but not a hydrolysis transition state analog ADP·VO$_4^{3-}$, support cohesin loading. The energy from nucleotide binding is sufficient to drive the DNA entry reaction into the cohesin ring. ATP hydrolysis, believed to be essential for in vivo cohesin loading, must serve a subsequent reaction step. These results provide molecular insights into cohesin function and open new experimental opportunities that the budding yeast model affords.

## Introduction

Cohesin, a ring-shaped multisubunit protein assembly conserved from yeast to humans, plays crucial roles in chromosome biology (Nasmyth & Haering, 2009; Peters & Nishiyama, 2012; Uhlmann, 2016). The complex is essential for sister-chromatid cohesion, as well as interphase and mitotic genome organization, transcriptional regulation, and DNA repair. Defects in human cohesin and its regulators are the cause for genetic developmental disorders, including Cornelia de Lange syndrome, Roberts syndrome, and Warsaw breakage syndrome. In addition, mutations in genes encoding cohesin subunits and regulators are frequent in cancer genomes (Losada, 2014).

The cohesin subunits Smc1 and Smc3 are characterized by a long stretch of flexible coiled coil, with an ABC family ATPase head domain at one end and a dimerization interface at the other. Dimerization at this interface, known as the "hinge," generates V-shaped Smc1-Smc3 heterodimers. The two ATPase head domains, in turn, afford ATP binding-dependent dimerization. A kleisin subunit, Scc1, bridges the ATPase heads to link them and reinforce their interaction. In addition, the HEAT repeat subunits Scc3 and Pds5, as well as Wapl, contact Scc1 and regulate cohesin function and dynamics. This ring-shaped cohesin complex assembly topologically embraces DNA to promote sister chromatid cohesion (Haering et al, 2008; Murayama et al, 2018).

Studies using budding yeast have offered insights into cohesin regulation and function. Cohesin loading onto chromosomes depends on the Scc2–Scc4 cohesin loader complex, which is recruited to nucleosome-free region (Ciosk et al, 2000; Lopez-Serra et al, 2014). From there, cohesin translocates along genes to reach its final places of residence at convergent transcriptional termination sites (Glynn et al, 2004; Lengronne et al, 2004; Ocampo-Hafalla et al, 2016). Cohesin loading occurs in late G1 phase, before initiation of DNA replication. However, cohesin loading onto chromosomes is not sufficient to generate sister chromatid cohesion, it requires a dedicated cohesion establishment reaction that takes place at the DNA replication fork (Uhlmann & Nasmyth, 1998; Skibbens et al, 1999; Tóth et al, 1999; Lengronne et al, 2006). Cohesion establishment involves the Eco1 acetyl transferase, which targets two conserved lysine residues on the Smc3 ATPase head (Ben-Shahar et al, 2008; Unal et al, 2008; Zhang et al, 2008). Smc3 acetylation is helped by several DNA replication proteins, including the Ctf18–RFC complex, the Mrc1-Tof1-Csm3 replication checkpoint complex, Ctf4, and Chl1 (Borges et al, 2013). Following DNA replication, sister chromatid cohesion is maintained until mitosis, when the protease separase is activated to cleave Scc1 and trigger chromosome segregation (Uhlmann et al, 2000).

Recent biochemical studies using fission yeast proteins have provided insights into how cohesin is loaded onto DNA (Murayama & Uhlmann, 2014, 2015). Cohesin loads topologically onto DNA in an ATP-dependent reaction that is facilitated by the cohesin loader. The fission yeast Mis4$^{Scc2}$-Ssl3$^{Scc4}$ cohesin loader complex contacts cohesin at several of its subunits and, in the presence of DNA, stimulates cohesin's ATPase. ATP, but not nonhydrolyzable ATP analogs ATP-γS or AMP-PNP, support cohesin loading, which led to

Chromosome Segregation Laboratory, The Francis Crick Institute, London, UK

Correspondence: frank.uhlmann@crick.ac.uk

the notion that ATP hydrolysis is required during the loading reaction. This idea is consistent with observations that Walker B motif mutations in cohesin's ATPase, that are thought to allow ATP binding but prevent ATP hydrolysis, block budding yeast cohesin loading onto chromosomes in vivo (Weitzer et al, 2003; Arumugam et al, 2003, 2006).

Fission yeast cohesin loading in vitro is promoted by the HEAT repeat-containing Mis4[Scc2] C-terminus; it does not require the Mis4[Scc2] N-terminus nor the Ssl3[Scc4] subunit that binds to it. The latter play their role during cohesin loading onto chromatin in vivo (Chao et al, 2015). Following topological loading onto DNA, fission yeast cohesin undergoes rapid one-dimensional diffusion along DNA that is constrained by DNA-binding proteins (Stigler et al, 2016). Similar diffusive sliding of topologically loaded vertebrate cohesin along DNA has been observed, although the contributions of ATP and of the human cohesin loader to cohesin loading remain less well characterized (Davidson et al, 2016; Kanke et al, 2016).

Despite our knowledge about the function of budding yeast cohesin in vivo, the reconstitution of its topological loading onto DNA in vitro has not yet been achieved. To investigate whether results obtained with the fission yeast proteins are more generally applicable and to further characterize the cohesin loading reaction, we have now purified budding yeast cohesin and its loader. As we observed with fission yeast proteins, the Scc2–Scc4 cohesin loader stimulates cohesin's ATPase and promotes topological in vitro cohesin loading onto DNA. Also in line with fission yeast, the nonhydrolyzable ATP analog ATP-γS fails to support cohesin loading. In contrast, we find that ADP in conjunction with ATP ground state mimicking phosphate analogs, ADP·BeF$_2$, ADP·BeF$_3^-$, and ADP·AlF$_x$, but not the ATP hydrolysis transition state mimetic ADP·VO$_4^{3-}$, efficiently promote topological cohesin loading. This observation reconciles previous results that cohesin ATPase Walker B mutations only mildly effect the in vitro cohesin loading efficiency. Together, this suggests that the energy from ATP binding is sufficient to fuel the DNA's entry reaction into the cohesin ring and that ATP hydrolysis serves a succeeding step during in vivo cohesin loading. The biochemical reconstitution of budding yeast cohesin loading onto DNA opens new experimental opportunities that this model organism affords, complementing approaches using fission yeast and vertebrate cohesin.

# Results

## Purification and biochemical characterization of budding yeast cohesin and its loader

We purified a budding yeast cohesin core tetramer complex, consisting of Smc1, Smc3, Scc1, and Scc3, following co-overexpression of the four subunits from galactose-inducible promoters in budding yeast (Figs 1A and S1A). The Scc2–Scc4 cohesin loader complex was similarly overexpressed and purified (Figs 1A and S1B). A gel mobility shift assay showed concentration-dependent DNA association of cohesin at a low salt concentration (Fig S2A). This DNA binding was independent of DNA topology and was equally observed with circular or linear DNA as the substrate, consistent with previous reports

(Losada & Hirano, 2001; Sakai et al, 2003; Murayama & Uhlmann, 2014). As expected, the Scc2–Scc4 complex also associated with DNA (Fig S2B).

Next, we characterized the ATPase activity of purified budding yeast cohesin. The cohesin complex by itself showed only little ATP hydrolysis, even in the presence of DNA. Addition of Scc2–Scc4 resulted in a substantial increase in the ATP hydrolysis rate (Fig 1B). This is qualitatively similar to the behavior of fission yeast cohesin and consistent with a recent report on ATP hydrolysis by budding yeast cohesin (Murayama & Uhlmann, 2014; Petela et al, 2018). In the presence of the cohesin loader and DNA, budding yeast cohesin hydrolyzed ~1 ATP per second, a rate that is six times faster than what was observed with fission yeast cohesin. We do not yet know whether this faster rate of ATP hydrolysis bears consequences on the function of budding yeast cohesin.

We also purified a cohesin complex containing glutamate to glutamine substitutions in the Walker B motifs of both Smc1 (E1158Q) and Smc3 (E1155Q) (Fig 1C). These substitutions are expected to allow ATP binding but to impede ATP hydrolysis (Lammens et al, 2004). Indeed, ATP hydrolysis by the resulting "EQ-cohesin" complex was substantially reduced, remaining only slightly above background levels even in the presence of the cohesin loader (Fig 1D). This documents that budding yeast cohesin contains a relatively fast ATPase, when stimulated by the cohesin loader.

## The cohesin loader promotes cohesin loading onto DNA

To study the loading of budding yeast cohesin onto DNA, we adapted an assay previously developed to study fission yeast cohesin (Murayama & Uhlmann, 2014) (Fig 2A). Cohesin and a circular plasmid DNA substrate were incubated in a low ionic strength buffer in the presence of ATP. Then, cohesin was immunoprecipitated and washed at a higher salt concentration to remove nontopologically bound DNA. Following the washes, cohesin-bound DNA was recovered and analyzed by gel electrophoresis. In the absence of the cohesin loader, around 10% of the input DNA bound to cohesin following 60 min of incubation. This fraction increased over time and reached 18% after 180 min. Scc2–Scc4 addition substantially accelerated cohesin loading and increased the amount of recovered DNA to around 25% of the input after an hour and close to 30% after 180 min (Fig 2B). The cohesin loader stimulated cohesin loading in a dose-dependent manner (Fig S2C). This suggests that cohesin, as previously seen with fission yeast proteins, can load onto DNA in an autonomous fashion, but that the cohesin loader facilitates this reaction.

A common feature of cohesin loading reactions, using budding yeast, fission yeast, or vertebrate proteins, is the requirement for relatively low ionic strength during the loading incubation (Murayama & Uhlmann, 2014; Davidson et al, 2016; Kanke et al, 2016). To ensure that a less than physiologic salt concentration did not cause cohesin complex dissociation or aggregation, we incubated cohesin under cohesin loading conditions and analyzed its oligomeric state before and after the incubation by size exclusion chromatography. This revealed that cohesin retains its elution characteristic as a single peak at the expected size for the tetrameric protein complex with an elongated shape (Fig S3). We do not currently know the reason for why in vitro cohesin loading is facilitated by low salt concentrations. A lower ionic strength might favor a conformation of cohesin, or

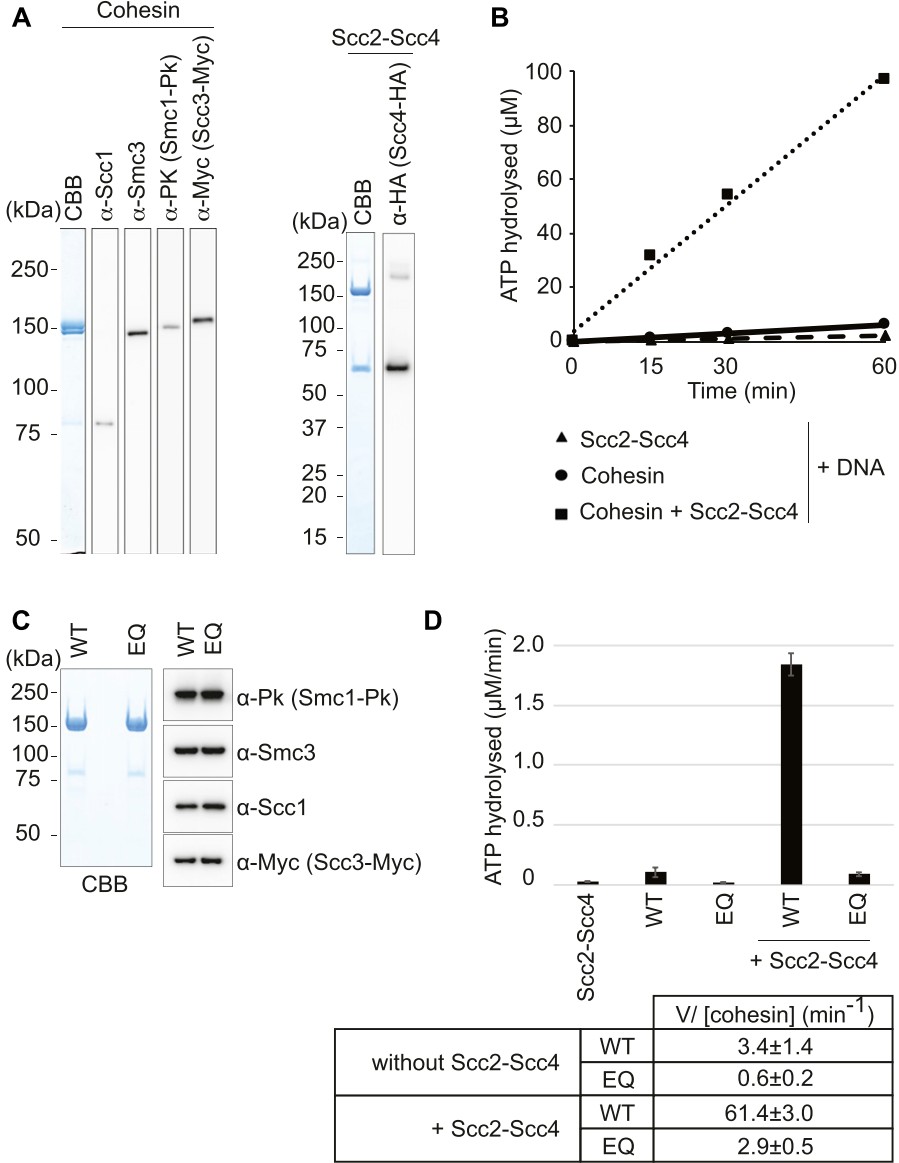

**Figure 1. Purification of budding yeast cohesin and its loader.**
**(A)** Purified budding yeast cohesin and cohesin loader were analyzed by SDS–PAGE, followed by Coomassie Blue staining (CBB) and immunoblotting with the indicated antibodies. **(B)** Time course analysis of ATP hydrolysis by cohesin in the presence of DNA, with or without the cohesin loader. **(C)** Purified cohesin and Walker B motif mutant EQ-cohesin were analyzed by SDS–PAGE, followed by Coomassie Blue staining and immunoblotting. **(D)** Comparison of the ATP-hydrolysis rates of wild type and EQ-cohesin, in the presence or absence of the cohesin loader. A reaction with the cohesin loader (Scc2–Scc4) but without cohesin served as a negative control. The mean values and standard deviations from three independent experiments are shown. Hydrolysis rates calculated per cohesin complex are listed.

| | | V/ [cohesin] (min$^{-1}$) |
|---|---|---|
| without Scc2-Scc4 | WT | 3.4±1.4 |
| | EQ | 0.6±0.2 |
| + Scc2-Scc4 | WT | 61.4±3.0 |
| | EQ | 2.9±0.5 |

interactions with the cohesin loader, that are conducive to the loading reaction.

### Functional modularity of the cohesin loader

To explore the previously observed functional modularity of the cohesin loader (Takahashi et al, 2008; Chao et al, 2015; Hinshaw et al, 2017), we also purified an Scc2 C-terminal fragment (Scc2C) encompassing amino acids 127–1,493 (Fig 2C). This includes an α-helical globular domain, as well as the hook-shaped C-terminal HEAT repeats, but lacks the Scc2 N-terminus to which Scc4 binds (Kikuchi et al, 2016; Chao et al, 2017). A gel mobility shift analysis showed that Scc2C binds to DNA in a manner indistinguishable from that of the Scc2–Scc4 complex (Fig S2B). This suggests that the DNA-binding activity of the cohesin loader is contained within Scc2C. In addition, Scc2C promoted ATP hydrolysis by cohesin and its topological loading

onto DNA as efficiently as the Scc2–Scc4 complex (Figs 2D and S2D). These results suggest that Scc4 and the Scc2 N-terminus are dispensable for cohesin loading onto DNA in vitro, as was observed with fission yeast proteins (Chao et al, 2015). The functional modularity, with Scc4 bound to the Scc2 N-terminus acting as a chromatin receptor and Scc2C catalyzing the loading reaction, emerges as a conserved aspect of the cohesin loader.

### Topological loading of budding yeast cohesin onto DNA

Cohesin is thought to perform its function on chromosomes by topologically entrapping DNA. We therefore investigated whether the budding yeast cohesin loader promotes topological loading of cohesin onto DNA. First, we compared DNAs of different topologies as substrates in the cohesin loading reaction. Supercoiled circular plasmid DNA, as well as relaxed or nicked circular DNA, served as

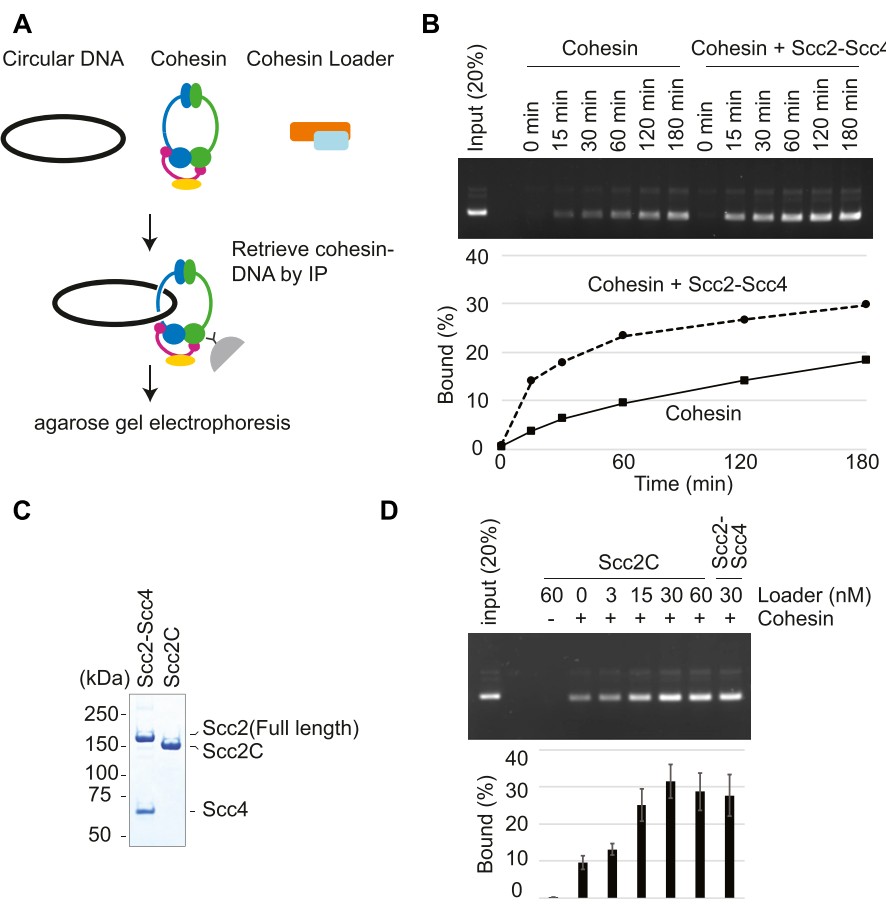

**Figure 2. Cohesin-loader−stimulated cohesin loading.**
**(A)** Schematic of the cohesin loading assay. Circular DNA and cohesin, with or without the cohesin loader, are incubated in the presence of ATP. Cohesin-DNA complex are retrieved by immunoprecipitation using an antibody against the Pk epitope tag on the Smc1 subunit. The recovered DNA is analyzed by agarose gel electrophoresis. **(B)** Gel image and quantification of a cohesin loading time course experiment in the presence or absence of the cohesin loader. **(C)** The Scc2−Scc4 complex was analyzed by SDS−PAGE and Coomassie Blue staining next to the Scc2C fragment. **(D)** Gel image and quantification of recovered DNA from the cohesin loading assay performed with the indicated concentration of Scc2C in comparison with the Scc2/Scc4 complex. Mean values and standard deviations from three independent experiments are shown.

equally efficient substrates in the loading reaction. In contrast, the linearized plasmid was not retained by cohesin (Fig S4A). This is consistent with the possibility that cohesin topologically embraces DNA during the loading reaction and that the topological nature of binding is required for cohesin to retain DNA during the washing steps.

To confirm topological binding of cohesin to DNA, we loaded cohesin onto a supercoiled plasmid DNA as the substrate. Following retrieval of cohesin–DNA complexes from the loading reaction, we linearized DNA with the restriction enzyme PstI or performed a control incubation without restriction enzyme (Fig 3A). Linearized DNA was released into the supernatant, whereas circular DNA remained bound to cohesin on the beads (Fig 3B). This was observed in reactions both with or without the cohesin loader. Thus, cohesin topologically embraces DNA in a reaction that is stimulated by the cohesin loader. Scc2C similarly stimulated the recovery of topologically bound DNA (Fig S4B). These observations corroborate the conclusion that cohesin topologically loads onto DNA in a reaction that is facilitated by an activity contained in the C-terminus of the Scc2 cohesin loader subunit. The reconstitution of topological budding yeast cohesin loading onto DNA in vitro opens the possibility to use engineered covalent subunit interface closures, designed in this organism (Gligoris et al, 2014), to further study the process.

Separase cleaves the cohesin subunit Scc1 during mitosis to dissociate cohesin from chromosomes and trigger anaphase

(Uhlmann et al, 2000). To investigate whether Scc1 cleavage releases topologically bound cohesin from DNA in our assay, we replaced one of the two separase-recognition sequences in Scc1 with two tandem tobacco-etch virus (TEV) protease recognition motifs. After loading of the modified cohesin complex onto DNA, TEV protease was added to half of the reaction and cohesin was retrieved (Fig 3C). Immunoblotting showed that Scc1 containing TEV recognition sites, but not wild-type Scc1, was efficiently cleaved by TEV protease (Fig 3D). TEV protease incubation did not affect DNA binding by wild-type cohesin, but resulted in DNA loss from TEV-cleavable cohesin. These results suggest that budding yeast cohesin is topologically loaded onto DNA in vitro in a way that makes it susceptible to DNA release by Scc1 cleavage, analogous to cohesin cleavage in anaphase.

## Low-level DNA loading without added ATP

The above cohesin loading reactions were all performed in the presence of ATP. We next explored the nucleotide requirements for cohesin loading. When we omitted ATP from the loading reaction, small amounts of DNA were still retrieved (Fig 4A). Furthermore, this portion of DNA was topologically bound to cohesin (Fig S5A). A low level of topological cohesin loading without added ATP was previously also observed with fission yeast and human cohesin (Murayama & Uhlmann, 2014; Davidson et al, 2016). One possibility is

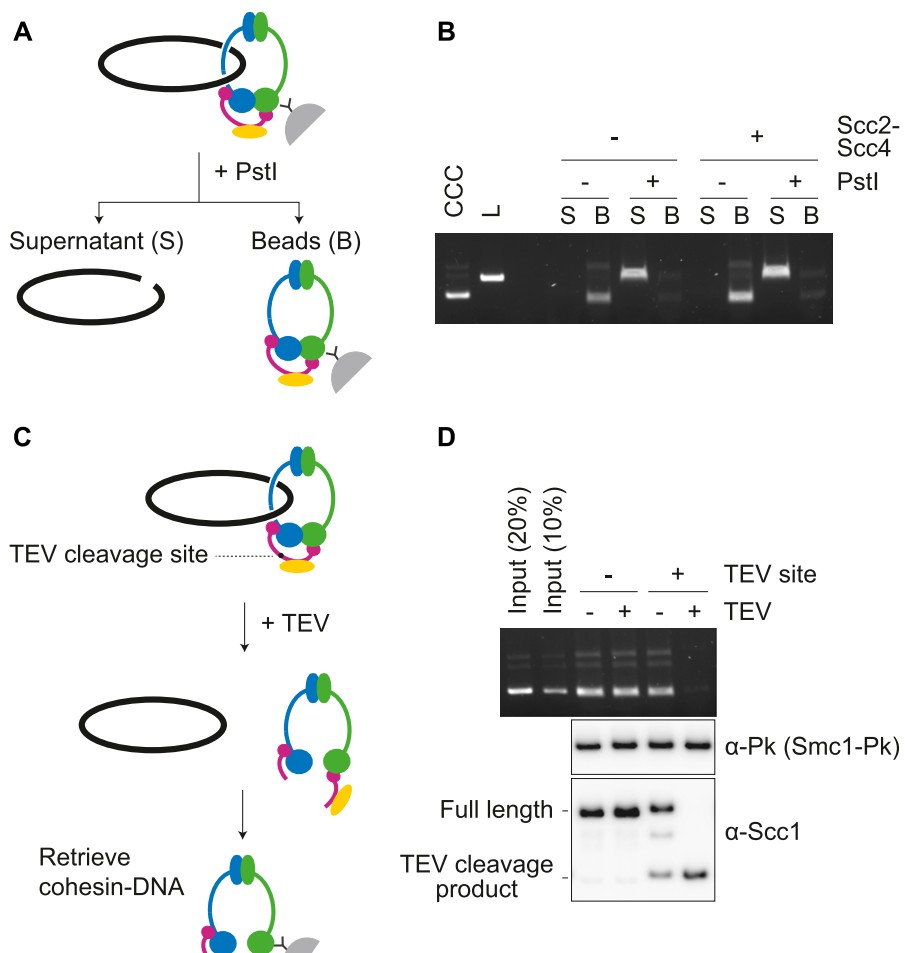

**Figure 3. Topological DNA embrace by the budding yeast cohesin ring.**
**(A)** Schematic of DNA release by DNA linearization. Immobilized cohesin-DNA complexes were incubated in the presence or absence of PstI. The supernatant fraction (S) and bead-bound fraction (B) were collected, and DNA in each fraction was analyzed by agarose gel electrophoresis. **(B)** Gel image of an experiment as outlined in (A). Cohesin loading was performed with or without Scc2–Scc4. Covalently closed circular (CCC) and linear (L) forms of the input DNA were included as a comparison. **(C)** Schematic of DNA release by cohesin cleavage. **(D)** Wild type and TEV protease (TEV)-cleavable cohesin were loaded onto DNA, then TEV protease was added to half of the reaction. Cohesin was retrieved and recovered DNA analyzed by agarose gel electrophoresis. Scc1 cleavage was monitored by immunoblotting. Note that TEV-cleavable cohesin was partially cleaved even without TEV addition. This could be due to similarities between the TEV and PreScission protease recognitions sites, the latter was used during the cohesin purification.

that a fraction of cohesin retained bound ATP during its purification. This fraction of cohesin might then be able to load onto DNA without the need for added ATP.

In reactions without added ATP, Scc2–Scc4 did not stimulate cohesin loading. On the contrary, the cohesin loader impeded loading (Fig 4A). Scc2C similarly limited cohesin loading in the absence of added ATP (Fig S5B). Loading inhibition by the cohesin loader, when no ATP is added, is specific to budding yeast and was not seen in the case of fission yeast cohesin (Murayama & Uhlmann, 2014). It could relate to a budding yeast-specific feature of the cohesin ATPase, namely that it is activated by the cohesin loader even in the absence of DNA (Petela et al, 2018). In this way, the cohesin loader might catalyze the depletion of copurified ATP, before cohesin had a chance to load onto DNA. This effect is not expected in the case of fission yeast cohesin, whose ATPase becomes active only when cohesin, the cohesin loader, and DNA come together.

### ATP binding, but not its hydrolysis, is required for in vitro cohesin loading

To address whether ATP must be hydrolyzed during cohesin loading, we used the ATP hydrolysis–defective EQ–cohesin complex

(Lammens et al, 2004; Arumugam et al, 2006; Hu et al, 2011). In fission yeast, an analogous Walker B motif mutant cohesin complex shows strongly reduced ATPase activity, but retains substantial topological DNA loading potential (Murayama & Uhlmann, 2015). Similarly, in the case of the budding yeast proteins, the loading efficiency of EQ-cohesin was comparable to that of wild-type cohesin (Fig 4B). DNA binding by EQ-cohesin was topological in nature (Fig S5C). Notably, in the absence of the cohesin loader, EQ-cohesin surpassed wild-type cohesin in its ability to load onto the DNA. This might be because EQ-cohesin shows greater stability on DNA following loading (Murayama & Uhlmann, 2015). The fact that Walker B motif mutant cohesin binds topologically to DNA suggests that ATP hydrolysis is not rate-limiting for cohesin loading. Rather, the loading reaction can take place without or with only minimal ATP hydrolysis.

When studying fission yeast cohesin loading onto DNA, we used nonhydrolyzable ATP analogs, ATP-γ-S and AMP-PNP. Neither of these supported cohesin loading. Furthermore, ATP-γ-S competed with ATP to inhibit cohesin loading (Murayama & Uhlmann, 2015). This implied that ATP-γ-S binds cohesin but does not support cohesin loading. We took this as a sign that ATP must be hydrolyzed during the loading reaction. On the other hand, a large sulfur replaces an

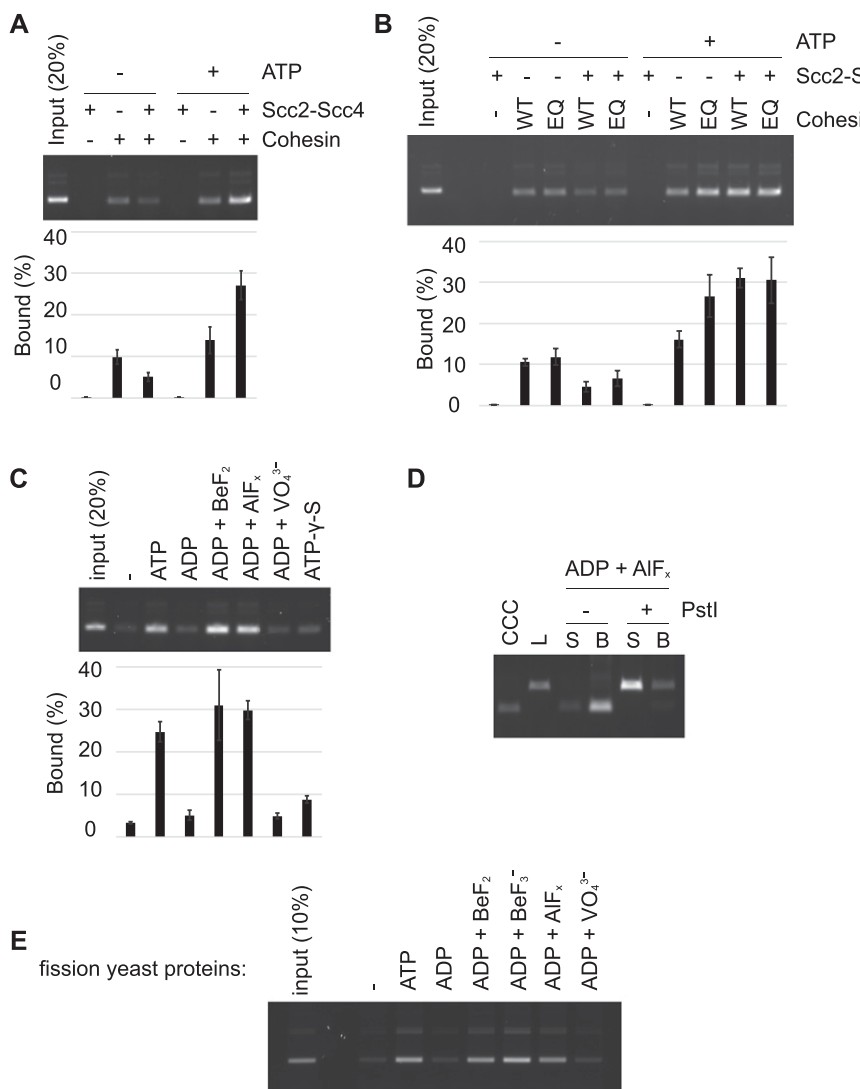

**Figure 4.    ATP binding, but not hydrolysis, is required for cohesin loading.**
**(A)** Gel image and quantification of recovered DNA from cohesin loading reactions performed with or without added ATP. **(B)** Gel image and quantification of recovered DNA from cohesin-loading reactions performed with wild-type or Walker B motif mutant EQ-cohesin. **(C)** An assay in which ATP and the indicated nucleotide derivatives were compared for their ability to support cohesin loading. The mean values and standard deviations from three independent experiments are shown in panels (A–C). **(D)** The topological nature of cohesin loading, supported by ADP·AlF$_x$, was analyzed following DNA linearization. **(E)** Fission yeast cohesin loading onto DNA was measured in the presence of ATP or the indicated nucleotide derivatives.

oxygen in ATP-γ-S, and the nitrogen in AMP-PNP introduces an angle into the otherwise colinear triphosphate. In addition to being hydrolysis-deficient, these ATP analogs might encounter a steric clash when entering cohesin's ATP binding site. ATP-γ-S and AMP-PNP might thus be imperfect mimetics for ATP binding. Consistent with this possibility, ATP-γ-S fails to promote Smc1-Smc3 head domain dimerization in a biochemical assay, whereas ATP supports dimerization of ATP hydrolysis-defective heads (Hu et al, 2011).

To explore the role of ATP binding and hydrolysis further, we prepared additional nonhydrolyzable ATP analogs that better mimic the geometry of ATP. These are ADPs in conjunction with the phosphate analogs beryllium fluoride (BeF$_2$), aluminum fluoride (AlF$_x$), and orthovanadate (VO$_4^{3-}$). The active form of BeF$_2$ in aqueous solution, BeF$_2$OH$^-$, shows a tetrahedral geometry, resembling the γ-phosphate of an ATP-bound ground state. AlF$_x$, obtained by combining AlCl$_3$ with NaF exists as a mixture of AlF$_3$, with a similar tetrahedral ground state geometry, and AlF$_4^-$. Crystal structures of an ABC family ATPase with these phosphate analogs show how AlF$_4^-$, as well as VO$_4^{3-}$, fit the trigonal bipyrimidal geometry of

a γ-phosphate during the transition state of hydrolysis (Combeau and Carlier, 1989; Oldham and Chen, 2011).

Strikingly, both ADP·BeF$_2$ and ADP·AlF$_x$ supported cohesin loading at a level equivalent or even greater than ATP (Fig 4C). In contrast, ADP, ADP·VO$_4^{3-}$, or ATP-γ-S did not support cohesin loading over what is observed in the absence of an added nucleotide. We used the same batch of ADP for all nucleotide preparations to avoid confounding effects due to possible ATP contamination in commercially obtained ADP. We further confirmed that DNA binding in the presence of ADP·AlF$_x$ was topological in nature (Fig 4D). These results demonstrate that ATP binding by cohesin is sufficient and that hydrolysis of bound ATP is not required for topological cohesin loading onto DNA. Furthermore, cohesin loading requires the presence of an ATP ground state mimetic.

Given our previous observations that ATP-γ-S and AMP-PNP were unable to support fission yeast cohesin loading, we revisited the fission yeast cohesin loading reaction by including the additional, nonhydrolyzable nucleotide mimetics. We performed cohesin loading reactions using the purified fission yeast cohesin complex and its Mis4$^{Scc2}$-Ssl3$^{Scc4}$ cohesin loader as previously described

(Murayama & Uhlmann, 2014). In addition to the above-listed phosphate analogs, we also prepared $BeF_3^-$ by combining $BeSO_4$ and NaF. Similar to what we observed with the budding yeast proteins, $ADP·BeF_2$, $ADP·BeF_3^-$, and $ADP·AlF_x$, but not ADP or $ADP·VO_4^{3-}$, supported cohesin loading (Fig 4E). This reveals a conserved requirement for ATP binding in its ground state geometry, which supports cohesin loading in the absence of ATP hydrolysis.

## Discussion

We have biochemically characterized cohesin loading with purified proteins from budding yeast. This has shown that many of the features observed with fission yeast proteins are similarly seen with those from budding yeast. Topological loading onto DNA is an activity intrinsic to the cohesin ring. It is facilitated by the Scc2–Scc4 cohesin loader complex, particularly its Scc2C module. ATP hydrolysis by cohesin is stimulated by the cohesin loader, but is not in fact required for cohesin's topological DNA embrace. Rather, the engagement of cohesin loader with ATP-bound cohesin is sufficient to achieve DNA entry into the cohesin ring. Budding and fission yeasts are evolutionarily distant, so we expect that these conserved features will be applicable to most eukaryotes. The nature of the conformational changes that ATP binding brings about, and how the cohesin loader facilitates them, are important topics for future research. Our observation that a nonhydrolyzable ATP analog must adhere to an ATP ground state geometry suggests that an ATP-bound state, possibly with the two ATPase heads firmly engaged, confers cohesin loading.

Although Walker B motif mutant cohesin loads efficiently onto DNA in vitro, cohesin loading in vivo is severely compromised by Walker B motif mutations (Arumugam et al, 2003; Weitzer et al, 2003). In particular, EQ-cohesin fails to reach stable chromosome association at centromeres and does not reach its final binding sites in the vicinity (Hu et al, 2011). The consequences of ATP hydrolysis and how this completes the in vivo cohesin loading reaction will be important to explore. While we were revising this study, Camdere et al, 2018 reported that $ADP·AlF_x$ stabilizes the interaction between cohesin and the cohesin loader. It could therefore be that ATP hydrolysis serves to release cohesin from the cohesin loader. This might be important to complete cohesin loading in the context of chromatin. It could also serve to regenerate free cohesin loader for additional loading cycles. The authors suggest that $ADP·AlF_x$ supports cohesin loading as an ATP hydrolysis transition state mimetic. Our further analysis of phosphate analogs suggests that it is likely that $ADP·AlF_3$, similar to $ADP·BeF_3^-$, but unlike $ADP·AlF_4^-$ or $ADP·VO_4^{3-}$, promoted cohesin loading by mimicking stably cohesin-bound ATP in its ground state.

Budding yeast is a widely used model organism for studying chromosome biology, covering many aspects that intersect with cohesin function. These include DNA replication, DNA repair, chromatin assembly, chromosome condensation, and transcriptional regulation. As an example, sister chromatid cohesion is established concomitantly with DNA replication and chromosome replication can now be studied using purified budding yeast proteins (Ticau et al, 2015; Yeeles et al, 2015). Similarly, budding yeast chromatin remodelers that

function during cohesin loading and during replication-coupled chromatin assembly are amenable to biochemical studies (Lopez-Serra et al, 2014; Lorch et al, 2014; Kurat et al, 2017). We expect that the ability to load budding yeast cohesin onto DNA in vitro will synergize with neighboring fields to enhance our molecular understanding of cohesin function in the wider context of chromosome biology.

## Materials and Methods

### Yeast cohesin and cohesin loader expression constructs

PCR amplified Smc1-encoding genomic DNA, fused to three tandem Pk epitopes at the C-terminus, was cloned under the control of the bidirectional *Saccharomyces cerevisiae* *GAL1-GAL10* promoter in the *GAL1* direction into the shuttle vector pRSII402 (*ADE2*). This plasmid also contained the budding yeast *GAL4* gene under the control of the *GAL10* promoter to improve galactose-induced protein expression, yielding pRSIISmc1-Gal4. Genomic DNA encoding Scc1, fused to two tandem protein A tags at the C-terminus, separated by a 3C protease recognition sequence, was cloned under the control of the *GAL1* promoter into YIplac204 (*TRP1*). TEV protease cleavable Scc1 was created by replacing the separase recognition sequence (SVEQGRR) with two TEV protease recognition sequences (ENLYFQ-GENLYFQG). Smc3-encoding genomic DNA was cloned into the same plasmid in the *GAL10* direction, yielding the plasmid YIpScc1-Smc3. Genomic DNA encoding Scc3, fused to a myc epitope tag at the C-terminus, was cloned under the control of the *GAL10* promoter into YIplac211 (*URA3*), yielding YIpScc3. The linearized pRSIISmc1-Gal4, YIpScc1-Smc3, and YIpScc3 plasmids were sequentially integrated into budding yeast (W303 background, *MAT***a** *pep4Δ::HIS3 wpl1Δ::LEU2 eco1Δ::KAN^R*) at their respective marker loci.

Scc2-encoding DNA, fused to two tandem protein A tags at the C-terminus, was cloned under the control of the *GAL1* promoter into pRSII402. The plasmid also contained the *GAL4* gene under the control of the *GAL10* promoter. Scc4-encoding DNA, fused to a triple HA epitope tag at the C-terminus, was cloned into YIplac204. The linearized pRSIIScc2-Gal4 and YIpScc4 were sequentially integrated into budding yeast at the respective marker loci, as above.

A Scc2 C-terminal fragment (Scc2C) encompassing amino acids 127–1,493, fused to a double HA epitope and protein A tag at the C-terminus, was cloned under the control of the *GAL1* promoter into pRSII402. The plasmid also contained the *GAL4* gene under the control of the *GAL10* promoter, yielding pRSIIScc2-Gal4. Linearized pRSIIScc2-Gal4 was integrated into budding yeast at the *ADE2* locus.

### Cohesin purification

Cells harboring the Smc1, Smc3, Scc1, and Scc3 expression constructs were grown in YP medium containing 2% raffinose as the carbon source to an optical density of 1.0 at 30°C. 2% galactose was then added to the culture to induce protein expression for further 2 h. Cells were collected by centrifugation, washed once with deionized water and resuspended in an equal volume of buffer A (50 mM Hepes-NaOH pH 7.5, 2 mM $MgCl_2$, 20% [vol/vol] glycerol, 0.5 mM Tris(2-carboxyethyl)phosphine hydrochloride [TCEP], 0.5 mM

Pefabloc [Sigma-Aldrich], and a protease inhibitor cocktail), containing 300 mM NaCl. The cell suspension was frozen in liquid nitrogen and broken in a freezer mill. The cell powder was thawed on ice, then two volumes of buffer A, containing 300 mM NaCl and RNase A (0.3 $\mu$g/ml final) was added. The lysates were clarified by centrifugation at 30,000 $g$ for 30 min at 4°C, then at 142,000 $g$ for 1 h. The clarified lysate was added to pre-equilibrated IgG agarose beads (Sigma-Aldrich) for 2 h in the presence of 1.25 U/ml benzonase. The resin was washed with buffer A containing 300 mM NaCl and incubated overnight in the same buffer containing PreScission protease (10 $\mu$g/ml final). The eluate was loaded onto a HiTrap Heparin HP column (GE Healthcare). The column was developed with a linear gradient from 300 mM to 1 M NaCl in buffer A. The peak fractions were pooled and loaded onto a Superose 6 10/300 GL gel filtration column (GE Healthcare) that was equilibrated and developed with buffer R (20 mM Tris–HCl pH 7.5, 150 mM NaCl, 10% glycerol, 0.5 mM TCEP). The peak fractions were concentrated by ultrafiltration. Cohesin containing a TEV protease cleavage site was purified using the same procedure.

### Purification of the cohesin loader

Cells harboring the Scc2 and Scc4 expression constructs were grown as above, but galactose induction of Scc2–Scc4 expression was for 1.5 h. Cells were collected by centrifugation, washed with deionized water, and resuspended in an equal volume of buffer B (50 mM Tris–HCl pH 8.0, 2.5 mM MgCl$_2$, 10% glycerol, 1 mM DTT, 0.5 mM PMSF and a protease inhibitor cocktail), containing 300 mM NaCl. The cell suspension was frozen in liquid nitrogen and broken in a freezer mill. The cell powder was thawed on ice, then two volumes of buffer B, containing 300 mM NaCl and RNase A (0.3 $\mu$g/ml final) was added. The lysates were clarified by centrifugation at 30,000 $g$ for 30 min at 4°C, then at 142,000 $g$ for 1 h. The clarified lysate was added to pre-equilibrated IgG agarose beads for 2 h in the presence of 1.25 U/ml benzonase. The resin was washed with buffer B containing 300 mM NaCl and incubated overnight in the same buffer containing 3C protease. The eluate was loaded onto a HiTrap Heparin HP column that was developed with a linear gradient from 300 mM to 1 M NaCl in buffer B. The peak fractions were pooled and loaded onto a Superdex 200 10/300 GL (GE Healthcare) gel filtration column that was equilibrated and developed in buffer R. The peak fractions were concentrated by ultrafiltration.

Purification of Scc2C followed essentially the same procedure, except that buffer B containing 10 mM NaCl was added to the eluate from the IgG agarose beads to adjust the salt concentration to 100 mM NaCl. This diluted eluate was loaded onto a HiTrap Heparin HP column that was developed with a linear gradient from 100 mM to 1 M NaCl in buffer B. The peak fractions were pooled and loaded onto a Superdex 200 10/300 GL gel filtration column. Scc2C-containing peak fractions were concentrated as described above.

### Electrophoretic gel mobility shift assay

Increasing concentrations of cohesin, the Scc2–Scc4 complex, or Scc2C were incubated for 30 min with 2.5 nM (molecules) of the indicated topologies of pBluescript KSII (+) at 29°C in 35 mM Tris–HCl pH 7.0, 20 mM NaCl, 0.5 mM MgCl$_2$, 13.3% glycerol, 0.5 mM ATP, 0.003% Tween-20, and 1 mM TCEP. The reactions were then separated on a 0.8% agarose/Tris-Acetate-EDTA buffer (TAE) gel by electrophoresis. DNA was detected by staining with GelRed (Biotium).

### ATPase assay

30 nM cohesin, 60 nM Scc2/4, and 3.3 nM pBluescript KS II (+) DNA were combined in 35 mM Tris–HCl pH 7.0, 20 mM NaCl, 0.5 mM MgCl$_2$, 13.3% glycerol, 0.003% Tween-20, and 1 mM TCEP. Reactions were initiated by the addition of 0.25 mM ATP, spiked with [$\gamma$-$^{32}$P]-ATP, and incubated at 29°C. Reaction aliquots were retrieved at 0, 15, 30, and 60 min and terminated by adding 125 mM EDTA. 1 $\mu$l of the reactions were spotted onto polyethylenimine cellulose F sheets (Merck) and separated by thin-layer chromatography using 0.75 M KH$_2$PO$_4$ (pH 3.4) as the mobile phase. The separated spots representing ATP and released inorganic phosphate were quantified using a Phosphorimager and Fiji software.

### Budding yeast in vitro cohesin loading assay

The standard reaction volume was 15 $\mu$l. 30 nM cohesin, 60 nM Scc2–Scc4, and 3.3 nM (molecules) pBluescript II KS(+) DNA were combined in 35 mM Tris–HCl pH 7.0, 20 mM NaCl, 0.5 mM MgCl$_2$, 13.3% glycerol, 0.5 mM ATP, 0.003% Tween, and 1 mM TCEP. Alternatively, 0.5 mM ADP, 0.5 mM ATP-$\gamma$-S, or 0.5 mM ADP supplemented with 2.5 mM BeF$_2$, 2.5 mM AlCl$_3$-10 mM NaF, or 2.5 mM Na$_3$VO$_4$ were included instead of ATP. The reactions were incubated at 29°C for 120 min if not otherwise stated. To stop the loading reactions, 500 $\mu$l of IP Buffer 1 (35 mM Tris–HCl pH 7.5, 100 mM NaCl, 10 mM EDTA, 5% glycerol, 0.35% Triton X-100) was added to the reaction mixture. $\alpha$-Pk antibody-coated protein A-conjugated magnetic beads were added and rocked at 4°C for 14 h. The beads were washed four times with IP Buffer 1 and then once with IP Buffer 2 (35 mM Tris–HCl pH 7.5, 100 mM NaCl, 0.1% Triton X-100). The beads were suspended in 12 $\mu$l of elution buffer (10 mM Tris–HCl pH 7.5, 1 mM EDTA, 50 mM NaCl, 0.75% SDS, 1 mg/ml protease K) and incubated at 37°C for 30 min. The recovered DNA was analyzed by 0.8% agarose gel electrophoresis in TAE buffer; the gel was stained with GelRed. Gel images were captured using an Amersham Imager 600 (GE Healthcare) or Gel Doc XR+ Documentation System (Bio-Rad); band intensities were quantified using Fiji.

In experiments that included linearization of cohesin-bound covalently closed circular DNA, the cohesin-bound DNA was retrieved by immunoprecipitation as described above. The beads were further washed with restriction enzyme buffer (35 mM Tris–HCl pH 7.5, 100 mM NaCl, 10 mM MgCl$_2$, 0.1% Triton X-100, 0.1 mg/ml BSA). The beads were incubated with PstI (20 U, New England Biolabs) in 12 $\mu$l restriction enzyme buffer at 10°C for 2 h. DNA in the supernatant and beads fractions were analyzed as described above.

Cleavage of engineered Scc1 by TEV protease was carried out at 16°C for 30 min using 2 units of AcTEV protease (Invitrogen) added to the reaction mixture following loading. Cohesin-bound DNA was analyzed as above.

## Fission yeast in vitro cohesin loading assay

Fission yeast cohesin tetramer (100 nM) and Mis4-Ssl3 (100 nM), purified as described (Murayama & Uhlmann, 2014) were mixed with circular pBluescript II KS(+) DNA (3.3 nM) in CL buffer (35 mM Tris–HCl pH 7.5, 25 mM NaCl, 1 mM $MgCl_2$, 15% [vol/vol] glycerol, 0.003% Tween 20, and 1 mM TECP). The reactions (15 $\mu$l) were initiated by the addition of 0.5 mM ATP, 0.5 mM ADP or 0.5 mM ADP, and 0.5 mM $BeF_2$, 0.5 mM $BeSO_4$-10 mM NaF, 0.5 mM $AlCl_3$-10 mM NaF, or 0.5 mM $Na_3VO_4$, and incubated at 32°C for 120 min. To terminate the reactions, 500 $\mu$l of CP1 buffer (35 mM Tris–HCl pH 7.5, 500 mM NaCl, 10 mM EDTA, 5% [vol/vol] glycerol, 0.35% Triton X-100 and 1 mM TCEP) was added. Pk antibody bound to protein A-conjugated magnetic beads were added to the mixture and rocked at 4°C overnight. The beads were washed three times with CP1 buffer and once with CP2 buffer (35 mM Tris–HCl pH 7.5, 50 mM NaCl, 10 mM EDTA, 5% [vol/vol] glycerol, 0.35% Triton X-100, and 1 mM TECP). Cohesin-bound DNA was eluted at 50°C for 20 min in 15 $\mu$l elution buffer (10 mM Tris–HCl pH 7.5, 1 mM EDTA, 50 mM NaCl, 0.75% SDS, and 1 mg/ml protease K). The recovered DNA was analyzed by 0.8% agarose gel electrophoresis in TAE buffer. The gel was stained with SYBR Gold (Thermo Fisher Scientific), and gel images were captured using a Typhoon FLA 9500 Imager (GE Healthcare).

### Antibodies

Mouse monoclonal antibodies against Smc3 and Scc1 were generously provided by K. Shirahige. Antibodies against the Pk (clone SV5) and myc (clone 9E10) epitopes were purchased from Bio-Rad.

# Supplementary Information

# Acknowledgements

The authors thank N Patel and A Alidoust from the Crick Fermentation Science Technology Platform for their support and members of our laboratory for discussions and critical reading of the manuscript. This work was supported by an EMBO Long Term Fellowship to M Minamino, the European Research Council (grant agreement no. 670412), and by The Francis Crick Institute, which receives its core funding from Cancer Research UK (FC001198), the UK Medical Research Council (FC001198), and the Wellcome Trust (FC001198).

## Author Contributions

M Minamino: conceptualization, data curation, formal analysis, investigation, methodology, and writing—original draft, review, and editing.
TL Higashi: conceptualization, data curation, investigation, and methodology.
C Bouchoux: data curation, investigation, and methodology.
F Uhlmann: conceptualization, formal analysis, and writing—original draft.

## Conflict of Interest Statement

The authors declare that they have no conflict of interest.

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
