## [Reviewer comments · Life Science Alliance]

Life Science Alliance

Topological in vitro loading of the budding yeast cohesin ring onto DNA

Masashi Minamino, Torahiko Higashi, Céline Bouchoux, and Frank Uhlmann

Corresponding author(s): Frank Uhlmann, The Francis Crick Institute

Review Timeline:

Submission Date:	2018-07-30
Editorial Decision:	2018-09-06
Revision Received:	2018-10-06
Editorial Decision:	2018-10-10
Revision Received:	2018-10-10
Accepted:	2018-10-10

Scientific Editor: Andrea Leibfried

Transaction Report:

DOI: 10.26508/lsa.201800143

September 6, 2018

Re: Life Science Alliance manuscript #LSA-2018-00143

Dr. Frank Uhlmann
The Francis Crick Institute
Chromosome Segregation Laboratory
1 Midland Road
London NW1 1AT
United Kingdom

Dear Dr. Uhlmann,

Thank you for submitting your manuscript entitled "Topological in vitro loading of the budding yeast cohesin ring onto DNA" to Life Science Alliance. The manuscript was assessed by expert reviewers, whose comments are appended to this letter.

As you will see from the reports, the referees appreciate that your work extends your earlier findings from fission yeast to budding yeast and that you provide the first demonstration that ATP hydrolysis per se is dispensable for cohesin loading in vitro. However, the referees also raise a number of concerns that we would like you to address in a revised version of the manuscript. From our side, point #3 from ref #1 is beyond the scope of the study and should be discussed only, while points #1 and #2 will in our view improve the conclusiveness of the work and should be included. The concerns from refs #2 and #3 can be addressed with additional clarification and a few control experiments to rule out contamination in the ADP vials.

Thank you for this interesting contribution to Life Science Alliance. We look forward to receiving

your revised manuscript.

Sincerely,

Andrea Leibfried PhD
Executive editor
Life Science Alliance

- A letter addressing the reviewers' comments point by point.
- An editable version of the final text (.DOC or .DOCX) is needed for copyediting (no PDFs).
- High-resolution figure, supplementary figure and video files uploaded as individual files: See our detailed guidelines for preparing your production-ready images, <http://life-science-alliance.org/authorguide>
- Summary blurb (enter in submission system): A short text summarizing in a single sentence the study (max. 200 characters including spaces). This text is used in conjunction with the titles of papers, hence should be informative and complementary to the title and running title. It should describe the context and significance of the findings for a general readership; it should be written in the present tense and refer to the work in the third person. Author names should not be mentioned.

B. MANUSCRIPT ORGANIZATION AND FORMATTING:

Full guidelines are available on our Instructions for Authors page, <http://life-science-alliance.org/authorguide>

Reviewer #1 (Comments to the Authors (Required)):

In their manuscript, Uhlmann and colleagues recapitulate, using budding yeast cohesin complexes, previous in vitro experiments that reported the loading of fission yeast cohesin onto DNA. As seen for the fission yeast complex, purified budding yeast condensin associates with different plasmid DNA substrates under low salt conditions, but retains binding only to circular DNAs after washing

with high salt, which suggests that the complex topologically associated with DNA. This loading reaction is stimulated by the cohesin loader Scc2/Scc4 or part of the Scc2 protein alone and reversed by DNA linearization or proteolytic cleavage of cohesin. ATP, as well as two ATP analogues that are thought to trap the transition state of the cohesin ATPase, promote loading of budding or fission yeast cohesin, which is similarly unaffected by a mutation that prevents ATP hydrolysis. The authors conclude that ATP hydrolysis is dispensable for the loading of cohesin onto DNA.

Although we appreciate the authors' work and find it very satisfying that they can reproduce key aspects of their previous work with a cohesin complex from a different yeast species, we have major concerns regarding both the novelty of the manuscript as well as the biological relevance of the assay used.

1. In its current state, this manuscript presents very limited novelty. Contrary to the authors' assertions, it has already been shown that not only fission yeast cohesin but also purified frog and human cohesin complexes can topologically load onto DNA *in vitro* (Kanke et al., EMBO J 2016; Davidson et al., EMBO J 2016). Moreover, the authors (Chao et al., NSMB 2017) and others (Petela et al., Mol Cell 2018) have previously purified budding yeast cohesin complexes for biochemical studies. Finally, the conclusion that ATP hydrolysis is not required for the loading reaction was already evident from previous work of the authors with fission yeast cohesin (Murayama & Uhlmann, Cell 2015, Figure 3), since a Walker B ("transition state") mutant cohesin complex failed to hydrolyze ATP but still loaded onto DNA.

2. The fact that the same mutant fails to load onto chromosomes *in vivo* (Srinivasan et al., Cell 2018) raises strong reservations about the biological relevance of the *in vitro* system used by the authors. Moreover, to achieve efficient (~30%) loading of cohesin onto DNA, the authors need to incubate cohesin with DNA for 120-180 min at 29 deg. C, a time span that corresponds to roughly two budding yeast cell cycles. If one molecule of cohesin hydrolyses one molecule of ATP per second under the conditions of the assay, one would need to assume that cohesin needs to perform many futile ATPase cycles before it succeeds in loading onto DNA. *In vivo*, the loading reaction would need to take place in a fraction of the time it takes *in vitro*, which raises doubts whether the reaction the authors observe in their assay makes biological sense. Finally, the loading reactions (like the gelshift assays) were performed at unphysiologically low salt concentrations, which further questions the specificity of these reactions.

One explanation for this incongruity might be that the observed association with DNA could be the result of protein aggregation, which is likely to occur over such a long time scale at 29 deg. C, especially at the very low ionic strengths conditions of the assay. To test for protein aggregation, the authors should rerun their cohesin complexes on size-exclusion chromatography after two hours incubation in the low salt reaction buffer at 29 deg. C. If this experiment confirms that the majority of cohesin complexes still elutes as a distinct single peak, this would greatly improve the interpretability of the experiments.

With these concerns in mind, we cannot recommend the manuscript for publication in Life Science Alliance. The work requires significant additional experimentation to substantiate the validity of the *in vitro* assay as well as the claims made in regard to the transition state of the ATPase being sufficient for topological entrapment of DNA.

A revised version should:

1. Exclude protein aggregation (see above).

2. Include formal proof that the assay indeed probes topological loading of cohesin onto DNA that recapitulates the in vivo loading of the complex. With the budding yeast cohesin complex now at hand, the authors should perform covalent crosslinking of the Smc1, Smc3 and Scc1 subunits to obtain clear evidence that DNA is located inside the ring (see Gligoris et al., Science 2014).

3. To substantiate that the transition state of the ATPase heads is sufficient for allowing DNA entrapment, the authors should determine binding affinities of the nucleotides to the ATPase heads and assess to which extent these nucleotides promote head dimerization. In an ideal case scenario, structures of Smc3/Smc1 complexed either with ATP- γ -S or ADP BeF₁₃/AlF₄ would explain why one nucleotide analogue can support loading while the other cannot. These experiments would clarify whether the "transition state" induced by either of these analogues corresponds to a true biochemical transition state in which the angle/bond length of the gamma-phosphate oxygens are altered on their way to hydrolysis.

Minor comments:

Figure 1A: The stoichiometry of the complex cannot be determined from this gel. The authors have previously accomplished better resolution (Chao et al., NSMB 2017; Figure 4), as have others for budding yeast cohesin (Petela et al., Mol Cell 2018; Figure 1). Proof of stoichiometry is especially relevant since fission yeast cohesin complexes purified by the authors yielded substoichiometric amounts of Psc3, which needed to be compensated for by supplementing Psc3 in the loading reactions.

Figures 1A, 1C and 2B: Coomassie gels and Western blots should be rendered with a non-zero background.

Figures 1B and D: Please fit the line/curve that was used to derive the hydrolysis rates rather than connect the data points. Error bars should be added to Figure 1D, which otherwise implies that the experiment had only been performed once.

Figures 2A and 3A: It is unclear whether DNA and cohesin are interlinked in the scheme.

Figure S1B: It would be useful if the authors could include the gelfiltration profiles to indicate whether some proteins also eluted at the void volume of the column. The gel also reveals an additional band at ~70 kDa, which might be a heat-shock protein. Did the authors observe co-purification of heat shock proteins by mass spec?

Reviewer #2 (Comments to the Authors (Required)):

This is a straightforward manuscript describing the purification and establishment of in vitro loading of cohesin from budding yeast onto DNA. Similar experiments have been described for pombe, so this piece of work is mostly replication of the previous results with proteins from budding yeast. The most exciting aspect of the work is that the loading reaction may only require ATP binding and not hydrolysis. However, I have some concern about this result (see below) that should be addressed. Issues to be addressed:

1. Figure 2-does the truncated form of Scc2 stimulate ATP hydrolysis?

2. Was the same vial of ADP used to prepare all nonhydrolyzable analogs? One concern is that

ADP can be contaminated with ATP, and if a different lot was used to prepare, for example, orthovanadate vs beryllium fluoride and aluminum fluoride, the results could be due to different levels of contamination. If true, the suggestion that ATP binding is sufficient for loading would be invalid and the novelty of the study would be diminished. The authors should mention something about reproducibility and what controls were done to eliminate this possibility.

3. If the concerns in part 2 can be addressed, then the authors should give a more detailed explanation of how these analogs "better mimic the geometry of the transition state." This is really the most exciting part of the paper and the authors should offer more explanation. For example, why doesn't the vanadate work, assuming of course that this one is not just the most pure (least amount of ATP) and that is why it doesn't work.

4. The loading inhibition by Scc2-Scc4 specific to budding yeast is surprising. Is there something about the recombinant protein that inhibits loading? What if this protein is added to the fission yeast reaction, does it still have this effect?

5. Do the authors care to speculate what the hydrolysis step is for?

Reviewer #3 (Comments to the Authors (Required)):

Short summary of the paper, including description of the advance offered to the field:

The previous work of the authors concerning the in-vitro loading of fission yeast cohesin onto DNA was a major breakthrough in the field. In the current manuscript they demonstrate the in-vitro loading of the budding yeast cohesin complex using the purified cohesin loader complex from budding yeast. This is a very important piece of work since a lot of general mechanisms for the cohesin complex have been established using budding yeast, yet the potential to perform in-vitro loading experiments was still missing. This allows now to validate results from fission yeast in another organism.

The second important observation of the authors is that in contrast to the non-hydrolysable ATP analogue gATPs, the transition state analogues ADP-ALF4- and ADP-BeF3- can support cohesin loading. This gives new insights into the cohesin-loading mechanisms and the role of ATP hydrolysis in there.

Support of the conclusions by the data:

The different experiment demonstrating the cohesin-loading by the budding yeast loader complex are of high quality and this reviewer has no doubt about the conclusions drawn.

In figure 4 C-E the authors present the DNA-binding capacity of budding and fission yeast cohesin in the presence of several ATP analogues. It would be important to show here quantitative analyses as in panels A and B. This would be important to understand whether the analogues regain full loading capacity as ATP. Actually the PCR bands look even stronger than for ATP, could the loading in the presence of the analogues be even more efficient? The authors use here ADP-BeF3, ADP-AIF4 (written ALF4 in the figures!) and ADP-VO4. Loading is detected for ADP-BeF3 and ADP-AIF4.

Looking through literature it seems that these analogues represent different stages of the ATP hydrolysis cycle and ADP-BeFx is more a ground-state mimic and ADP-AIFx a transition-state mimic (eg. Ponomarev, FEBS, 1995 and Chen ... Nixon, Structure, 2007). Therefore it would be important to quantitate these experiments and eventually support them with other available analogues. Over longer terms a co-crystal structure would be of great interest, although this is clearly beyond what can be asked for the revision of this manuscript.

Minor issues:

Figure 1

Restructure the table below panel D, it is unreadable and add the error bars to the graph.

Figure 2

Add the error bars to the graph in panel B.

We would like to thank the three reviewers for their interest in our study and for their insightful and largely constructive comments. Please find below a point-by-point response how we have used these comments to improve the manuscript.

Reviewer 1 appreciates our work but raises major concerns. In the order that these were raised:

1. We fully agree with the reviewer that numerous others have made contributions to our biochemical knowledge of cohesin function in various organisms. Indeed, all the examples mentioned by the reviewer are cited and discussed in our manuscript.

On the other hand, we respectfully disagree with the reviewer that our manuscript presents limited novelty. Confirmation of various functional aspects of fission yeast cohesin and its loader, using proteins from the evolutionarily distant budding yeast, is by no means trivial. Beyond that, the discovery that certain non-hydrolyzable ATP analogs, but not others, allow topological cohesin loading came unexpected. We expanded this part of our study during the revision, see below, to offer new insight into cohesin function.

2. The reviewer is right that the overall assay incubation time, of typically 2 hours, is similar in length to a budding yeast cell cycle at the same temperature. Most of the *in vitro* loading happens in the first hour, which is comparable to the time it takes for *in vivo* cohesin loading (see e.g. Lengronne et al. 2006, Figure 6A). It is therefore incorrect that “*in vivo*, the loading reaction would need to take place in a fraction of the time it takes *in vitro*”.

Further concerns are raised about the low ionic strength during the cohesin loading incubation, which might have led to artefactual protein aggregation. We can assure the reviewer that this is not the case, as detailed below in response to specific point 1.

Specific points:

1. As suggested by the reviewer, we have analyzed the oligomeric state of cohesin before and after incubation under conditions of cohesin loading. Size exclusion chromatography revealed that cohesin elutes in a single peak, at its expected position, both before and after the incubation. Therefore, cohesin remains a soluble and stable protein complex during the loading reaction. This control is included in the revised manuscript as a new Supplementary Figure S3.

2. The reviewer is right that the establishment of *in vitro* cohesin loading with budding yeast proteins opens the possibility to engineer interface crosslinks, following the approach of Gligoris et al. 2014. This is a major task, involving redesign of our expression constructs to include cysteines, which goes beyond the scope of practical revisions to our manuscript. This said, we are aware that the Nasmyth lab, who are championing the crosslinking approach, have performed this experiment with the expected outcome. We emphasize that our current manuscript confirms the topological nature of DNA binding by both DNA linearization and TEV cleavage of cohesin. It is unclear to us whether any possible ‘pseudo-topological’ binding must necessarily be sensitive to protein denaturation?

3. The reviewer asks for further details regarding nucleotide binding to the ATPase heads. Crystal structures of the Smc1-Smc3 head heterodimer bound to the different nucleotides would indeed be revealing, but go beyond the scope of our current work. Instead, in the revised manuscript, we discuss a crystallographic study of the bacterial MalK ABC transporter ATPase, bound to various ATP analogs (Oldham and Chen 2011). This study, together with a biochemical analysis of the state of these analogs when bound to actin and tubulin (Combeau and Carlier 1989), prompted us to extend our own analysis and revisit our conclusions. This revealed that ADP with phosphate analogs that mimic the ATP ground state (BeF_2OH^- and BeF_3^-) but not a hydrolysis transition state (VO_4^{3-}) support cohesin loading. Aluminum fluoride, in turn, exists as a mixture of AlF_3 and AlF_4^- (AlF_x), representing ground and transition state, respectively. The ground state analog AlF_3 may well be responsible for cohesin loading in a reaction with AlF_x . It therefore emerges that the ATP-bound ground state, possibly with stably engaged ATPase heads, promotes cohesin loading. These conclusions are documented in a revised Figure 4 and the accompanying text.

Minor points:

Figure 1A. We have now included a better resolved gel image. Though Coomassie Blue staining is not suitable to assess subunit stoichiometry. Rather, during size exclusion chromatography, the cohesin tetramer elutes distinctly earlier compared to a trimer lacking Scc3. We are therefore confident that the majority of the cohesin complexes used in our reaction contain Scc3.

Figures 1A, 1C and 2B. We have rendered the gel images such that background is visible. There is, however, very little background in ECL Western blot images captured with the Amersham Imager 600. We are happy to supply original image files, as appropriate.

Figure 1B and D. We have repeated the experiment shown in Figure 1D and have added error bars.

The schemes in Figures 2A and 3A have been improved according to the reviewers suggestion.

Figure S1 now includes the gel filtration profiles. As the reviewer rightly expects, the additional band at 70 kDa represents members of the Hsp70 family that were identified by mass spectrometry. This information was added to the figure legend.

Reviewer 2 considers this to be 'a straightforward manuscript', but makes several suggestions.

1. The truncated form of Scc2 indeed stimulates ATP hydrolysis. An experiment to document this has been included in the revised manuscript as a new panel in supplementary Figure S2D.

2. The nucleotide analogs are always freshly reconstituted in each assay, using ADP from the same vial in each reaction. In addition, the revised Figure 4C now reports the means and standard deviations from three independent experiments.

3. In the revised manuscript, we discuss a crystallographic study of the bacterial MalK ABC transporter ATPase, bound to various ATP analogs (Oldham and Chen 2011). This study, together with a biochemical analysis of the state of these analogs when bound to actin and tubulin (Combeau and Carlier 1989), prompted us to extend our own analysis and revisit our conclusions. This revealed that ADP with phosphate analogs that mimic the ATP ground state (BeF_2OH^- and BeF_3^-) but not a hydrolysis transition state (VO_4^{3-}) support cohesin loading. Aluminum fluoride, in turn, exists as a mixture of AlF_3 and AlF_4^- (AlF_x), representing ground and transition state, respectively. The ground state analog AlF_3 may well be responsible for cohesin loading in a reaction with AlF_x . It therefore emerges that the ATP-bound ground state, possibly with stably engaged ATPase heads, promotes cohesin loading. These conclusions are documented in a revised Figure 4E and the accompanying text.

4. The inhibition of cohesin loading by the Scc2-Scc4 cohesin loader, in the absence of added ATP, is indeed specific to budding yeast and is not seen when using fission yeast proteins. A budding yeast-specific feature of the cohesin ATPase, namely that it is activated by the cohesin loader even in the absence of DNA (Petela et al, 2018), could explain this difference. The cohesin loader might catalyze the depletion of copurified ATP, before cohesin had a chance to load onto DNA. This effect is not expected in case of fission yeast cohesin, whose ATPase becomes active only when cohesin, the cohesin loader and DNA come together.

5. A recent study from the Koshland lab (epub ahead of print) reports that cohesin and the cohesin loader interact more strongly in the presence of AlF_x , when compared with ATP. One possibility therefore is that ATP hydrolysis releases cohesin from the loader. This might be important to complete cohesin loading in the context of chromatin and could serve to regenerate free cohesin loader for additional loading cycles. We include these considerations in our revised discussion.

Reviewer 3 finds that ‘the different experiments demonstrating the cohesin-loading by the budding yeast loader complex are of high quality’, but has several comments:

In response to the reviewers valid concern about quantification of the results shown in Figure 4C, we have repeated this experiment and now report the means and standard deviations from three independent experiments. This confirms that ADP in conjunction with certain phosphate analogs promote cohesin loading at least equally efficiently as ATP. It is conceivable that DNA-cohesin complexes formed with non-hydrolysable ATP are more stable than those formed with ATP, owing to subsequent ATP hydrolysis-dependent cohesin unloading.

In the revised manuscript, we discuss a crystallographic study of the bacterial MalK ABC transporter ATPase, bound to various ATP analogs (Oldham and Chen 2011). This study, together with a biochemical analysis of the state of these analogs when bound to actin and tubulin (Combeau and Carlier 1989), prompted us to extend our own analysis and revisit our conclusions. This revealed that ADP with phosphate analogs that mimic the ATP ground state (BeF_2OH^- and BeF_3^-) but not a hydrolysis transition state (VO_4^{3-}) support cohesin loading. Aluminum fluoride, in turn, exists as a mixture of AlF_3 and AlF_4^- (AlF_x), representing ground and transition state, respectively. The ground state analog AlF_3 may well be

responsible for cohesin loading in a reaction with AIF_x. It therefore emerges that the ATP-bound ground state, possibly with stably engaged ATPase heads, promotes cohesin loading. These conclusions are documented in a revised Figure 4 and the accompanying text.

Minor issues:

Figure 1. We repeated the experiment shown in panel 1D and now report the means and standard deviation from three independent experiments.

Figure 2. The timecourse analysis is already a compilation of multiple measurements, so we did not add error bars to each individual measurement. Instead we have repeated the experiments shown in Figures 1D, 4C and in the new Figure S2D, so that we now show means and standard deviations for all assays that report single values as an experimental outcome.

October 10, 2018

RE: Life Science Alliance Manuscript #LSA-2018-00143R

Dr. Frank Uhlmann
The Francis Crick Institute
Chromosome Segregation Laboratory
1 Midland Road
London NW1 1AT
United Kingdom

Dear Dr. Uhlmann,

Thank you for submitting your revised manuscript entitled "Topological in vitro loading of the budding yeast cohesin ring onto DNA". Previous reviewer #2 re-assessed this version and now supports publication of your work. We also appreciate the introduced changes and would thus be happy to publish your paper in Life Science Alliance pending final revisions necessary to meet our formatting guidelines.

- please add a callout to Fig2C in your manuscript text

A. FINAL FILES:

-- High-resolution figure, supplementary figure and video files uploaded as individual files: See our detailed guidelines for preparing your production-ready images, <http://life-science-alliance.org/authorguide>

B. MANUSCRIPT ORGANIZATION AND FORMATTING:

Full guidelines are available on our Instructions for Authors page, <http://life-science-alliance.org/authorguide>

Sincerely,

Andrea Leibfried, PhD
Executive Editor
Life Science Alliance
Meyerohofstr. 1
69117 Heidelberg, Germany
t +49 6221 8891 502
e a.leibfried@life-science-alliance.org
www.life-science-alliance.org

Reviewer #2 (Comments to the Authors (Required)):

I am satisfied with the revisions.

October 10, 2018

RE: Life Science Alliance Manuscript #LSA-2018-00143RR

Dr. Frank Uhlmann
The Francis Crick Institute
Chromosome Segregation Laboratory
1 Midland Road
London NW1 1AT
United Kingdom

Dear Dr. Uhlmann,

Thank you for submitting your Research Article entitled "Topological in vitro loading of the budding yeast cohesin ring onto DNA". It is a pleasure to let you know that your manuscript is now accepted for publication in Life Science Alliance. Congratulations on this interesting work.

The final published version of your manuscript will be deposited by us to PubMed Central (PMC) as soon as we are allowed to do so, the application for PMC indexing has been filed. You may be eligible to also deposit your Life Science Alliance article in PMC or PMC Europe yourself, which will then allow others to find out about your work by Pubmed searches right away. Such author-initiated deposition is possible/mandated for work funded by eg NIH, HHMI, ERC, MRC, Cancer Research UK, Telethon, EMBL.

Please also see:

<https://www.ncbi.nlm.nih.gov/pmc/about/authorms/>

<https://europepmc.org/Help#howsubsmanu>

*****IMPORTANT:** If you will be unreachable at any time, please provide us with the email address of an alternate author. Failure to respond to routine queries may lead to unavoidable delays in publication.*******

DISTRIBUTION OF MATERIALS:

Again, congratulations on a very nice paper. I hope you found the review process to be constructive and are pleased with how the manuscript was handled editorially. We look forward to future exciting submissions from your lab.

Sincerely,
